# Examining a Genomic Test in Predicting Extended Endocrine Benefit and Recurrence Risk in a Diverse Breast Cancer Population

**DOI:** 10.3390/curroncol32100537

**Published:** 2025-09-25

**Authors:** Ho Hyun Lee, Nicholas Siu-Li, Ian Pagano, Jami Aya Fukui

**Affiliations:** 1Department of Medicine, John A. Burns School of Medicine, University of Hawai‘i, Honolulu, HI 96813, USA; 2Translational and Clinical Research, University of Hawaii Cancer Center, Honolulu, HI 96813, USA

**Keywords:** breast cancer, breast cancer index, extended endocrine benefit, hormone receptor-positive breast cancer, recurrence risk, race/ethnicity

## Abstract

The Breast Cancer Index is a molecular test that helps providers decide if patients should continue hormone therapy after the standard first five years to reduce cancer recurrence. However, little is known about how race and ethnicity may influence the results of this test. In this study, we analyzed the Breast Cancer Index scores from 159 women with early-stage hormone receptor-positive breast cancer to assess racial and ethnic differences. Japanese and other Asian/Pacific Islander patients were less likely to be classified as high risk for recurrence compared with Caucasian patients. There were no major racial or ethnic differences in predicting who would benefit from extended hormone therapy, supporting the test’s applicability across diverse racial and ethnic populations. These findings suggest potential variations in tumor biology or other contributing factors, underscoring the need for further research.

## 1. Introduction

The mainstay of care for early-stage hormone receptor-positive (HR+) breast cancer is adjuvant endocrine therapy for at least five years. Although the prognosis of early-stage HR+ breast cancer has improved with current therapies, more than a quarter of early HR+ breast cancers relapse [1]. This persistent risk of recurrence has led clinicians to strongly consider extending endocrine therapy beyond the first 5 years to reduce the risk of late recurrence. Prior studies have shown that extending endocrine therapy beyond the first 5 years reduces the risk of late distant recurrence [2,3]. The Breast Cancer Index (BCI) is a gene expression assay that evaluates the activity of 11 genes, including HOXB13 and IL17BR, to provide both a prognostic value and a prediction of benefit from extending endocrine therapy after the first 5 years. It is indicated for women diagnosed with HR+, lymph node-negative (LN−) or LN+ early-stage breast cancer. BCI is now a well-established, effective biomarker with both predictive and prognostic value, even in comparison to other biomarkers [1,4,5,6,7,8,9]. It has also been shown to decrease decision conflict and anxiety among patients and promote their adherence to extended endocrine therapy (EET), particularly in patients with a predicted high likelihood of benefit [10,11].

Despite the important role that BCI plays in clinical decision-making, there are currently no studies that characterize differences in BCI scores by race and their significance. The current study was aimed at evaluating potential racial and ethnic differences in late distant recurrence risk and predictive benefit of EET as indicated by BCI scores. This would help to strengthen the role of BCI in a diverse population of early-stage HR+ breast cancer patients.

## 2. Materials and Methods

Our retrospective chart review analyzed 159 women who were diagnosed with early-stage HR+ breast cancer between August 2020 and August 2023 and received treatment at oncology clinics within a major hospital system in Hawaii. Patients self-identified as Caucasian, Filipino, Japanese, Native Hawaiian, Other Asian/Pacific Islander (Other Asian PI), or Other. Inclusion criteria were an early-stage HR+ breast cancer diagnosis, an available BCI score, and a reported race. Exclusion criteria included metastatic breast cancer or triple-negative breast cancer diagnosis, absence of a BCI score, or lack of reported race. Tumor characteristics examined included tumor size (cm), grade (1–3), histology (ductal, lobular, or mixed), lymph node status, receptor status (ER/PR/HER2), Oncotype DX recurrence score, and laterality of cancer (right or left).

BCI prediction of EET benefit was reported as “yes” or “no”. Late distant recurrence risk was reported as a percentage and further classified into three categories for interpretability: low (<5%), mid (5–9%), and high (≥10%).

Statistical analyses were performed using SAS 9.4 (SAS Institute Inc., Cary, NC, USA). Univariable and multivariable logistic regression analyses were conducted to evaluate associations between race, BCI late distant recurrence risk (five to ten years from diagnosis), and tumor features with the outcome variables of BCI prediction of benefit and BCI recurrence risk. For the endocrine benefit model, binary logistic regression was used to model the probability of a “yes” outcome. For the BCI recurrence risk model, multinomial logistic regression was performed with low risk as the reference. Predictor variables were tumor size (0.1–0.9 cm, 1–1.9 cm, ≥2 cm), tumor grade (1, 2, 3), estrogen receptor status (0–49%, 50–100%), progesterone receptor status (0–49%, 50–100%), HER2 status (positive, negative), BCI risk (low, mid, high), and race. Variable categories were collapsed in analyzing BCI recurrence risk when small cell sizes prevented model convergence. Missing data in our analysis required for regression analyses, such as Oncotype DX recurrence scores, were handled by listwise deletion. We selected specific cutoff points for tumor size, grade, and receptor status based on clinical relevance and natural distribution patterns within our dataset. When possible, cutoff points reflected whole numbers or midpoints in clinical practice (e.g., 50% ER/PR). Both univariable (unadjusted) and multivariable (adjusted) models were run, and odds ratios with *p*-values were reported.

## 3. Results

We found differences in endocrine benefit prediction based on tumor size, tumor grade, lymph node status, histology, receptor status, BCI risk stratification, Oncotype DX score, laterality, and race (see Table 1).

Most patients in our study had a 1.0–1.9 cm tumor size (44.0%) and tumor grade 2 (46.5%). Study participants had predominantly negative lymph node status (76.7%) and ductal type (86.8%). HER2 receptor status was negative in the majority of patients (60.4%). Close to half of our patients were classified as high-risk for late distant recurrence (54.7%). For race and ethnicity, 43 patients (27.0%) self-identified as Caucasian, 18 (11.3%) as Filipino, 47 (29.6%) as Japanese, 23 (14.5%) as Native Hawaiian, 19 (11.9%) as Other Asian/Pacific Islander, and 9 (5.7%) as Other. There were no racial or ethnic differences in the prediction for benefit with EET. Patients with tumor grade 3 (OR = 7.64, *p* = 0.008), higher PR positive receptor status (OR = 2.85, *p* = 0.04), and invasive lobular histology (OR = 4.52, *p* = 0.02) were more likely to be predicted to benefit from EET compared with those with ductal histology. In comparison to those with low BCI risk (<5%), there were significantly greater odds of having a score predicting benefit among those who had a distant recurrence risk of 5–9% (OR = 5.84, *p* = 0.02) and ≥10% (OR = 4.16, *p* = 0.03). Univariable analysis showed similar trends except for PR receptor status (see Appendix A). The distribution of BCI-predicted endocrine benefit by race/ethnicity and the odds ratios for BCI endocrine benefit prediction by each predictor are shown in Figure 1 and Figure 2, respectively.

BCI categorized 48 patients (30.2%) as being at low, 24 (15.1%) at mid, and 87 (54.7%) at high risk of late distant recurrence. In multivariable analysis of BCI recurrence risk, patients with older age (60–69), positive endocrine therapy prediction, and higher tumor grade were associated with a higher risk for late distant recurrence. Positive HER2 and lymph node status were associated with a decreased likelihood of high distant recurrence risk (see Table 2). A similar trend was observed in univariate analysis, except for Oncotype DX recurrence score (see Appendix A).

Notably, Japanese patients had significantly lower odds of having a high risk for distant recurrence (OR = 0.05, *p* = 0.0002) compared to Caucasians. Other Asian/Pacific Islanders also showed reduced odds of high distant recurrence risk (OR = 0.09, *p* = 0.02), albeit to a lesser extent than the Japanese patients. Figure 3 shows the distribution of BCI recurrence risk scores by race/ethnicity. Further statistical clarity was achieved for Japanese and other Asian/Pacific Islander women in multivariable analysis. The odds ratios for mid-risk and high-risk recurrence by each predictor are shown in Figure 4 and Figure 5, respectively.

## 4. Discussion

This study shows that BCI can become impactful in clinical decision-making when considering racial and ethnic differences in scores and interpretations. Developing predictive biomarkers for breast cancer that can inform decisions to continue endocrine treatment has been a significant challenge due to the concerning rate of distant recurrence, which can exceed 25% after five years of adjuvant endocrine therapy [1]. This phenomenon has burdened providers with the task of more accurately assessing recurrence risk and promoting early intervention with extended adjuvant endocrine therapy for eligible individuals [6].

BCI has been serving as a well-established tool to guide them in deciding whether to initiate EET while estimating the risk of distant late recurrence [1,4,5,6,7,8,9,12,13,14]. It provides both prognostic and predictive information for patients with early-stage, HR-positive breast cancer. It is a molecular assay that examines the expression of 11 genes: seven genes associated with tumor proliferation and estrogen signaling (the Molecular Grade Index and HOXB13/IL17BR ratio) and four reference genes. The test provides an individualized genomic-based estimate of recurrence risk and endocrine responsiveness, identifying patients with low overall risk of distant recurrence over ten years, as well as late risk between five and ten years in patients who remain recurrence-free at year 5. This information can help weigh the potential benefits of EET against the risks of treatment-related toxicities and complications.

The predictive component of BCI has been validated in multiple prospective randomized cohorts, demonstrating that a higher HOXB13/IL17BR ratio is significantly associated with greater responsiveness to endocrine therapy [5,6]. The BCI has also been validated to stratify patients into low- versus high-risk categories for distant recurrence based on a combination of the H/I ratio and Molecular Grade Index [5]. A threshold of 10% risk of distant recurrence at 10 years is commonly used by several breast prognostic tests to define low risk [15,16]. However, no formally established thresholds exist in the literature to define percentage cutoffs for BCI risk categories. In this paper, we determined low (<5%), intermediate (5–9%), or high (≥10%) risk groups for clinical interpretability.

Despite extensive validation of BCI as a prognostic and predictive biomarker, prior studies have not examined the potential impact of a patient’s racial or ethnic profile on the results of BCI. In this study, patients of various races and ethnicities were identified, and their BCI results, predictions of benefits, and recurrence risks were compared.

Regarding the prediction of benefit with EET, our findings indicate no significant racial or ethnic differences. This is an important finding as it suggests that BCI’s predictive value is consistent across racial and ethnic groups, supporting its applicability in diverse populations of women with early-stage HR+ breast cancer. Lower BCI risk scores were also associated with increased odds of predicted benefit from extended endocrine therapy. However, this association is inherently driven by the BCI algorithm itself and does not reflect patient-level clinical or biological implications in our dataset.

Racial and ethnic disparities become evident in the analysis of late distant recurrence risk as predicted by the BCI test. Although not based on actual clinical outcomes, Japanese and other Asian/Pacific Islander patients were more likely to be characterized as low risk. This finding for Japanese patients aligns with favorable breast cancer outcomes observed in Japanese and Caucasian women according to the Hawaii Tumor Registry 2014–2018. Specifically, 17% of Japanese women were diagnosed at advanced stages compared to 33% of those from other racial or ethnic groups [17]. This suggests potential variations in tumor features or other genetic and environmental factors affecting distant recurrence risk among different racial groups, including Japanese and possibly other Asian/Pacific Islanders.

Some of these variations or factors may favor a lower distant recurrence risk among Japanese women. An extensive epidemiological study, utilizing 18 databases from Surveillance, Epidemiology, and End Results (SEER) spanning from 1975 to 2016, was conducted to characterize various features and mortality of breast cancer in Asian women in the United States. This study revealed that Japanese women exhibited several distinctively advantageous traits compared to other Asian groups: they were more likely to present with the lowest rate of stage IV disease, a higher propensity for hormone receptor positivity, an older median age at diagnosis, a lower rate of mastectomy and chemotherapy, and a higher rate of adjuvant radiation therapy within the entire Asian group [18]. When juxtaposed with both non-Hispanic White and other Asian subgroups, Japanese patients still demonstrated the highest rate of non-metastatic and node-negative disease, the lowest rate of HER2+ disease, the highest 10-year cancer-specific survival rate, and the highest proportion of being insured, thus better access to healthcare [18]. Hence, Japanese women’s protective factors that shield them from distant recurrence at the time of diagnosis could be attributed to multiple factors, encompassing both biological and social dimensions.

Other Asian/Pacific Islanders were also predicted by BCI to be less likely classified as high risk for late distant recurrence, albeit to a lesser extent than Japanese (OR = 0.07, *p* = 0.01). A limited number of studies have addressed the outcomes of breast cancer among these populations, let alone characterized their genetic and environmental differences in comparison to other racial or ethnic groups. One study concluded that Asian/Pacific Islander women overall had better breast cancer mortality rates compared to non-Hispanic White women, with some variability depending on specific national origins [19]. Therefore, similar to Japanese patients, Asian/Pacific Islander patients may also be presumed to have a unique set of genetic predispositions that make them advantageous in breast cancer outcomes, including distant recurrence. Furthermore, studies tend to aggregate many Asian/Pacific Islander subgroups into a single category, rendering it difficult to gain insight unique to each group. It is also essential to note the scarcity of research specifically focused on less predominant Asian ethnicities, such as Asian Indian, Pakistani, Cambodian, Hmong, or Laotian, as well as Pacific Islanders [18,20,21]. Our study was able to separate a few distinctive racial/ethnic groups, such as Filipinos and Native Hawaiians, and examine their characteristics compared to others. These findings are still with limitations, however, due to the small sample size, which leaves limited flexibility to disaggregate our patients further. Studies may reach different conclusions depending on the degree of disaggregation of patient populations each study achieves. Clinicians using BCI should therefore be cognizant of these racial and ethnic implications and strive to adopt more personalized treatment approaches.

Breast cancer outcomes are associated with various prognostic factors, including patient age, type of treatment received, hormone receptor and HER2 receptor status, lymph nodes, tumor grade, and tumor size [22]. Yet, less is known about how the histology of breast cancer influences the response to endocrine therapy or distant recurrence. Generally, lobular histology, as well as lymph node positive disease and HER2 status, are associated with a higher risk for late recurrence [23,24]. However, invasive lobular carcinoma is commonly known to be positive for hormone receptors, rendering it an ideal candidate for endocrine therapy [25,26]. Our result may be interpreted along the same line of thought as it demonstrated that patients with lobular cancer type are likely to benefit from EET. Reduced risk for distant recurrence in patients with positive HER2 and lymph node status is contradictory to the consensus, however, potentially due to our small sample size.

Our study is limited by a relatively small sample size and the lack of representation from certain racial groups, such as Hispanic and Black populations, which are known to experience disparities in breast cancer outcomes. Some racial/ethnic groups, such as Filipino (*n* = 18), had small subgroup sizes, limiting statistical power and the generalizability of our findings. The small sample size, along with the absence of survival analysis with actual patient outcomes over time, limits our ability to validate BCI scores across different racial and ethnic groups.

Additionally, this study lacks a detailed discussion of potential confounding factors that may have influenced recurrence risk among Japanese and Other Asian PI groups. These factors may include social determinants of health such as socioeconomic status, insurance coverage, access to care, and treatment adherence, as well as clinical variables like menopausal status and the duration of initial endocrine therapy. The information regarding other therapies patients may have received, including neoadjuvant therapy, other adjuvant therapies, radiotherapy, or ovarian function suppression, was also not available, as these data were not routinely captured in the chart records. Thus, the impact of these social and clinical factors on overall outcomes remains unknown.

These limitations may restrict the generalizability of the results. Future studies should aim to expand the sample size with adequate representation from all racial and ethnic groups and incorporate adjustments for established breast cancer risk factors to provide a more comprehensive understanding of racial and ethnic disparities in recurrence risk and BCI-predicted endocrine benefit.

## 5. Conclusions

Our findings highlight the utility of BCI’s predictive and prognostic scores across racial and ethnic groups among patients with early-stage HR+ breast cancer. Notably, Japanese and other Asian/Pacific Islander women had significantly decreased odds of having a score predicting high risk for late distant recurrence compared to Caucasian women. This provides unique insight into diverse racial and ethnic groups that are often studied as a single entity. BCI was also shown to be consistent in predicting benefits from EET regardless of race or ethnicity. Nevertheless, our sample size was relatively small, and detailed treatment data were lacking. Future studies should aim to expand sample size, ensure adequate racial and ethnic representation, and adjust for established prognostic factors to further refine the utility of the BCI as a tool to effectively guide hormone therapy recommendations for diverse patient populations with early-stage HR+ breast cancer.

## Figures and Tables

**Figure 1 curroncol-32-00537-f001:**
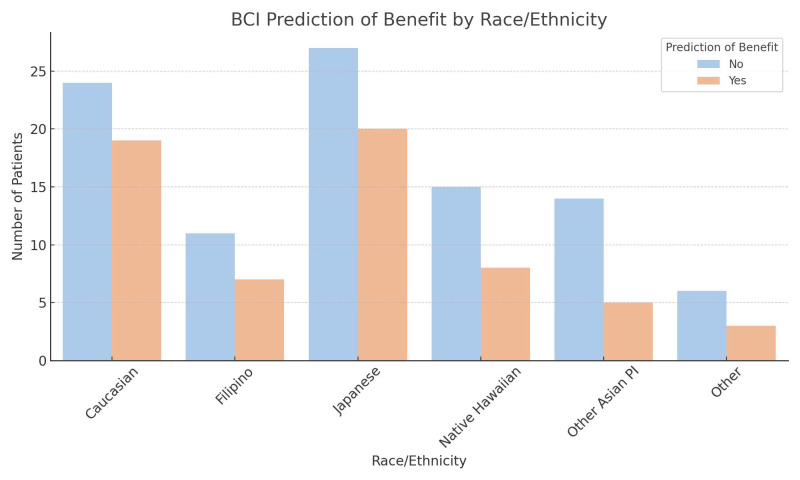
Distribution of BCI-predicted endocrine benefit across racial and ethnic groups.

**Figure 2 curroncol-32-00537-f002:**
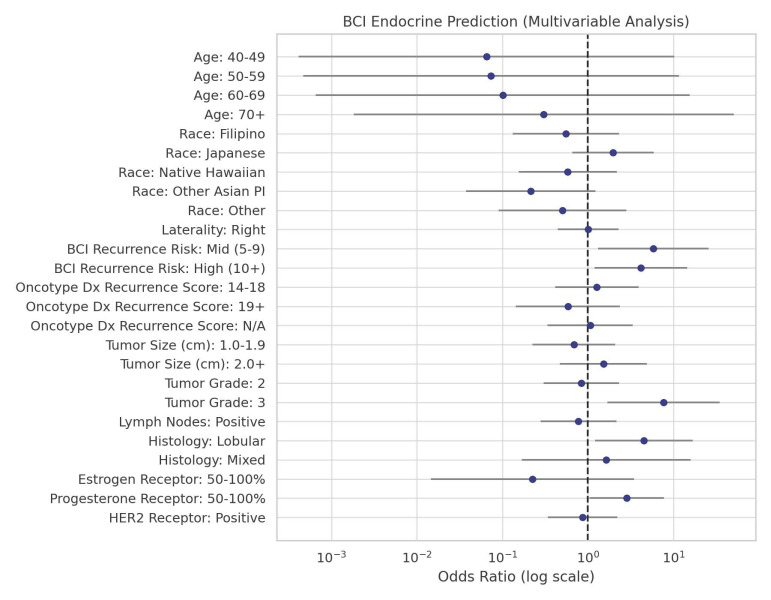
Odds ratios from multivariable analysis of predictors of BCI endocrine benefit. Effect sizes are shown as dots, with 95% confidence intervals represented by horizontal lines.

**Figure 3 curroncol-32-00537-f003:**
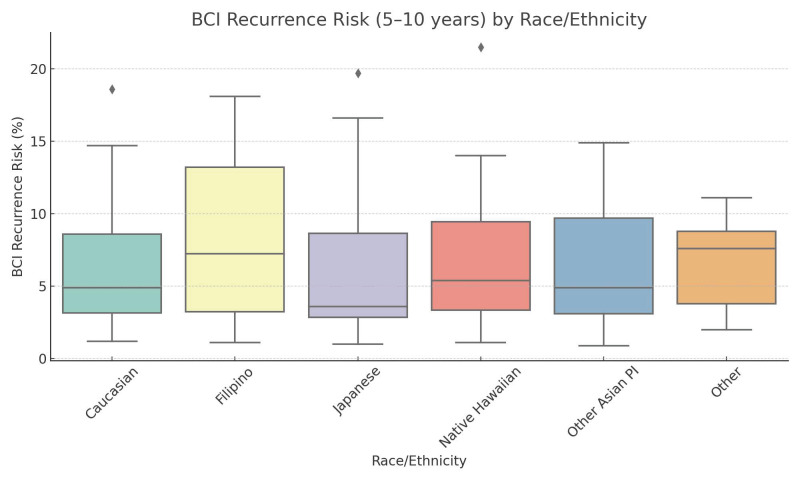
Distribution of BCI recurrence risk scores across racial and ethnic groups. Diamonds indicate outliers.

**Figure 4 curroncol-32-00537-f004:**
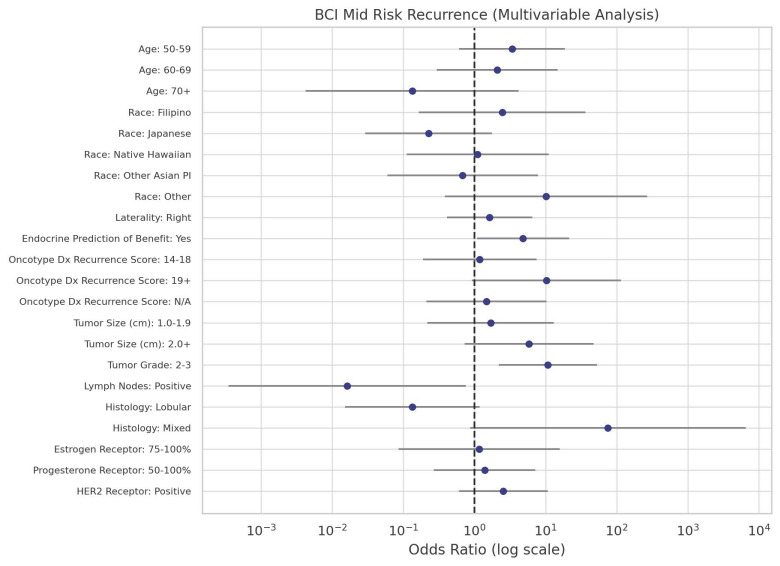
Odds ratios from multivariable analysis of predictors of BCI mid-risk recurrence. Effect sizes are shown as dots, with 95% confidence intervals represented by horizontal lines.

**Figure 5 curroncol-32-00537-f005:**
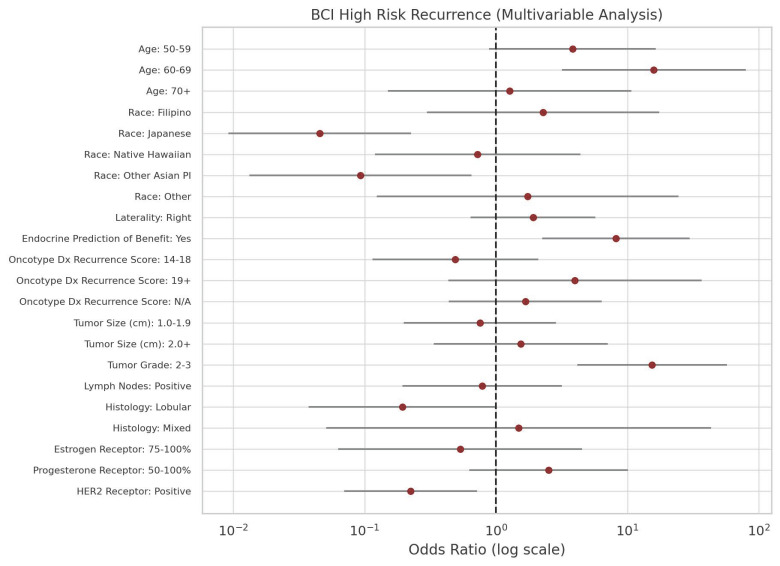
Odds ratios from multivariable analysis of predictors of BCI high-risk recurrence. Effect sizes are shown as dots, with 95% confidence intervals represented by horizontal lines.

**Table 1 curroncol-32-00537-t001:** Endocrine Prediction.

Variable		Total (*n* = 159)		Multivariable (EP YES)				
		*n*	Col%	%	OR	LCL	UCL	*p*
Age	30–39	4	2.5	75.0	1.00			
	40–49	40	25.2	16.3	0.07	0.00	10.27	0.29
	50–59	52	32.7	18.1	0.07	0.00	11.58	0.31
	60–69	47	29.6	23.2	0.10	0.00	15.55	0.37
	70+	16	10.1	47.7	0.30	0.00	51.03	0.65
Race	Caucasian	43	27.0	44.2	1.00			
	Filipino	18	11.3	30.5	0.55	0.13	2.32	0.42
	Japanese	47	29.6	60.8	1.96	0.65	5.86	0.23
	Native Hawaiian	23	14.5	31.5	0.58	0.16	2.17	0.42
	Other Asian PI	19	11.9	14.5	0.21	0.04	1.22	0.08
	Other	9	5.7	28.5	0.50	0.09	2.82	0.44
Laterality	Left	77	48.4	40.3	1.00			
	Right	82	51.6	40.4	1.01	0.45	2.28	0.99
BCI Recurrence Risk	Low (0–4)	48	30.2	18.8	1.00			
	Mid (5–9)	24	15.1	57.4	5.84	1.32	25.85	0.02
	High (10+)	87	54.7	49.0	4.16	1.19	14.46	0.03
Recurrence Score	0–13	40	25.2	32.5	1.00			
	14–18	37	23.3	38.0	1.27	0.41	3.93	0.67
	19+	28	17.6	21.9	0.58	0.14	2.38	0.45
	N/A	54	34.0	33.8	1.06	0.34	3.33	0.92
Tumor Size (cm)	0.0–0.9	38	23.9	34.2	1.00			
	1.0–1.9	70	44.0	26.3	0.69	0.23	2.09	0.51
	2.0+	51	32.1	44.1	1.52	0.47	4.90	0.49
Tumor Grade	1	55	34.6	27.3	1.00			
	2	74	46.5	23.9	0.84	0.30	2.32	0.73
	3	30	18.9	74.1	7.64	1.68	34.68	0.008
Lymph Nodes	Negative	122	76.7	41.0	1.00			
	Positive	37	23.3	35.0	0.77	0.28	2.15	0.62
Histology	Ductal	138	86.8	37.0	1.00			
	Lobular	17	10.7	72.6	4.52	1.21	16.85	0.02
	Mixed	4	2.5	48.9	1.63	0.17	15.82	0.67
ER	0–49	5	3.1	80.0	1.00			
	50–100	154	96.9	47.3	0.22	0.01	3.48	0.29
PR	0–49	47	29.6	36.2	1.00			
	50–100	112	70.4	61.7	2.85	1.04	7.79	0.04
HER2	Negative	96	60.4	38.5	1.00			
	Positive	63	39.6	35.2	0.86	0.34	2.20	0.76

**Table 2 curroncol-32-00537-t002:** Multivariable BCI Recurrence Risk.

Variable		Low		Mid			High		
		%	OR	%	OR	*p*	%	OR	*p*
Age	30–49	38.6	1.00	20.5	1.00		40.9	1.00	
	50–59	14.6	1.00	26.1	3.37	0.16	59.3	3.83	0.07
	60–69	5.3	1.00	5.8	2.09	0.46	88.9	15.92	0.0008
	70+	41.4	1.00	2.9	0.13	0.25	55.7	1.27	0.83
Race	Caucasian	20.9	1.00	9.3	1.00		69.8	1.00	
	Filipino	10.3	1.00	11.3	2.46	0.51	78.4	2.28	0.43
	Japanese	79.9	1.00	8.1	0.23	0.16	12.1	0.05	0.0002
	Native Hawaiian	25.6	1.00	12.6	1.11	0.93	61.8	0.73	0.73
	Other Asian PI	62.0	1.00	18.8	0.68	0.76	19.2	0.09	0.02
	Other	8.8	1.00	39.9	10.15	0.17	51.3	1.74	0.68
Laterality	Left	36.4	1.00	14.3	1.00		49.4	1.00	
	Right	23.6	1.00	15.2	1.64	0.49	61.3	1.92	0.24
Prediction	No	40.2	1.00	14.4	1.00		45.4	1.00	
	Yes	8.3	1.00	14.4	4.81	0.04	77.3	8.21	0.001
Recurrence Score	0–13	35.0	1.00	10.0	1.00		55.0	1.00	
	14–18	47.4	1.00	16.1	1.19	0.85	36.5	0.49	0.34
	19+	9.8	1.00	28.8	10.29	0.06	61.4	3.99	0.22
	N/A	24.7	1.00	10.4	1.48	0.69	64.9	1.67	0.45
Tumor Size (cm)	0.0–0.9	28.9	1.00	7.9	1.00		63.2	1.00	
	1.0–1.9	32.1	1.00	14.9	1.69	0.61	53.0	0.76	0.68
	2.0 +	16.8	1.00	26.8	5.87	0.10	56.4	1.54	0.58
Tumor Grade	1	56.4	1.00	9.1	1.00		34.5	1.00	
	2–3	8.2	1.00	14.3	10.79	0.003	77.6	15.45	<0.0001
Lymph Nodes	Negative	28.7	1.00	18.9	1.00		52.5	1.00	
	Positive	40.9	1.00	0.4	0.02	0.04	58.7	0.78	0.73
Histology	Ductal	29.7	1.00	15.2	1.00		55.1	1.00	
	Lobular	70.0	1.00	4.8	0.13	0.07	25.2	0.19	0.05
	Mixed	2.4	1.00	91.1	75.33	0.06	6.5	1.49	0.82
ER	0–74	14.3	1.00	14.3	1.00		71.4	1.00	
	75–100	20.7	1.00	24.1	1.16	0.91	55.2	0.53	0.57
PR	0–49	25.5	1.00	19.1	1.00		55.3	1.00	
	50–100	13.3	1.00	14.0	1.39	0.69	72.7	2.51	0.19
HER2	Negative	25.0	1.00	7.3	1.00		67.7	1.00	
	Positive	42.5	1.00	31.6	2.55	0.20	25.8	0.22	0.01

## Data Availability

The datasets supporting the conclusions of this article are included within the article and its Appendix A.

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
