# Peer review of "Examining a Genomic Test in Predicting Extended Endocrine Benefit and Recurrence Risk in a Diverse Breast Cancer Population"

_curroncol, 2025, doi:10.3390/curroncol32100537_

Round 1

Reviewer 1 Report (Previous Reviewer 1)

Comments and Suggestions for Authors

The authors have addressed the reviewers’ suggestions well. However, several issues remain that should be resolved before the manuscript is suitable for publication:

  1. This manuscript does not include a survival analysis. The recurrence risk was estimated based on the BCI platform alone. Therefore, the conclusion that “no racial/ethnic differences in EET benefit suggest BCI’s universal applicability” is not fully supported and may be overstated.
  2. To validate the BCI score across various racial and ethnic groups, a survival analysis with an adequate follow-up period should ideally be conducted. If this is not feasible, the authors should consider softening the conclusion to reflect the limitations of the current analysis.
  3. Line 166: It cannot be definitively stated that BCI’s predictive value is consistent across different racial and ethnic groups without supporting survival data.
  4. Lines 171 and 195: It should be clearly stated that the distant recurrence risk is predicted by the BCI test, not based on actual clinical outcomes.

Author Response

Reviewer 2 Report (New Reviewer)

Comments and Suggestions for Authors

The authors analyzed the benefit of BCI in predicting extended endocrine therapy in diverse breast cancer patient populations.

Please find below some thoughts for this manuscript:

  1. Detail the low and high recurrence risk of BCI as to explain for non-oncologists these categories. Please provide references for all your recommendations during text.
  2. Discuss the new adjuvant option of CDK4/6 inhibitors for high-risk early BC patients. The authors should specify that the benefit of testing BCI for extended adjuvant endocrine therapy is unknown in patients who had ovarian function suppression, CDK4/6 inhibitor addition to adjuvant endocrine therapy.
  3. Please revise the Methods section and reorganize better, meaning present step-by-step what did you do. The explanations about thresholds should not be provided here or explained better. Do not mix methods of choosing patients with statistics. Where were the patients treated: oncological institute? Tertiary hospital? Multiple centers? It is important to detail, because the sample size is small for a breast cancer retrospective study.
  4. Please refine “BCI’s prediction of benefit was reported as “yes” or “no” while late distant recurrence risk was given as a percentage. Distant recurrence risk was classified as low (<5%), mid (5-9%), or high (≥10%).” And later you discuss about Oncotype Dx. The methods section is confusing and the reader can not follow how the authors conducted their research.
  5. Early BC HR positive, HER2 negative is classified as Luminal A, B. If you put also in your analysis HER2pozitive, this cohort has another type of recurrence risk.
  6. The authors didn’t mention ki67…stage… other therapies, such as neo/adjuvant chemotherapy, adjuvant radiotherapy…
  7. Please revise the tables, there is too much information and it is difficult to follow it.
  8. Please reorganize the Discussion section and follow the cohort risks, your objectives and debate point by point.
  9. The authors should revise the text, as it seems it was written by many people and not revised in a final form.

Round 2

Reviewer 1 Report (Previous Reviewer 1)

Comments and Suggestions for Authors

The authors have well revised the manuscript.

This manuscript is a resubmission of an earlier submission. The following is a list of the peer review reports and author responses from that submission.

Round 1

Reviewer 1 Report

Comments and Suggestions for Authors

 The authors present valuable data that include various racial populations in the Hawaiian region. This study highlights the unique racial characteristics of the Asian/Pacific population and provides important insights regarding the diverse racial composition of Hawaii. In this multiethnic cross-sectional study, we can get an insight into extended endocrine therapy for breast cancer patients in the Asian/Pacific region. Although the conclusions are somewhat limited, the authors clearly acknowledge these limitations and successfully identify an association between Japanese patients and favorable outcomes. There are several points that should be addressed before the manuscript is published

Major Comments

  1. In the methodology section, the authors should specify the period during which patients were enrolled. Additionally, the inclusion and exclusion criteria for study enrollment should be clearly described. The authors should also explain the relatively small sample size (n = 159). Did the study include all patients who underwent BCI scoring?
  2. There are three supplementary tables, but no tables in the main manuscript. These supplementary tables should be incorporated into the main text as standard tables.

Minor Comments

  1. Were Japanese patients older than individuals from other populations?
  2. There is inconsistency in the classification of distant recurrence risk:
    • Line 59: "mid (5–10%)"
    • Line 88: "5–9%"
    • Line 95: "mid (5–10%)"
  3. Why did the authors describe the association between the BCI recurrence score and BCI’s prediction of benefit? This relationship is determined by the intrinsic algorithm of the BCI test itself, not by patient characteristics.
  4. In line 177, the phrase “hormone and ER/PR/HER2 receptor status” should be revised. "hormone status" refers to ER and PR.

Author Response

In the methodology section, the authors should specify the period during which patients were enrolled. Additionally, the inclusion and exclusion criteria for study enrollment should be clearly described. The authors should also explain the relatively small sample size (n = 159). Did the study include all patients who underwent BCI scoring?

- Thank you for your feedback. We have now specified the enrollment period in the Materials and Methods: 159 patients were identified between August 2020 and August 2023. We included patients who met the following inclusion criteria: early-stage hormone receptor-positive breast cancer diagnosis, availability of BCI score, and reported race/ethnicity. The exclusion criteria were: diagnosis of metastatic or triple-negative breast cancer, absence of a BCI score, or missing race/ethnicity information. All 159 patients were included in our study. These inclusion and exclusion criteria are now clearly stated in the Materials and Methods section. Some racial/ethnic groups, such as Filipino (n = 18), had small subgroup sizes, limiting statistical power and generalizability. We now acknowledge this as a key limitation in the Discussion section.

There are three supplementary tables, but no tables in the main manuscript. These supplementary tables should be incorporated into the main text as standard tables.

  • The supplementary tables have now been included as part of the main text.

Were Japanese patients older than individuals from other populations?

  • Thank you for your question. Japanese patients in our study had a mean age of 57 years (range: 31–75), which was comparable to other racial subgroups. Some groups were older, as White patients had a mean age of 58 years (range: 30–80). Across all groups, age distributions were similar, and no notable differences in age patterns were observed that would meaningfully influence our findings.

There is inconsistency in the classification of distant recurrence risk:

         Line 59: "mid (5–10%)"

         Line 88: "5–9%"

         Line 95: "mid (5–10%)"

  • Thank you for pointing this out. We have corrected the classification of mid-risk distant recurrence throughout the manuscript to consistently reflect 5–9%, in alignment with the thresholds used in our analysis.

Why did the authors describe the association between the BCI recurrence score and BCI’s prediction of benefit? This relationship is determined by the intrinsic algorithm of the BCI test itself, not by patient characteristics.

  • Thank you for this observation. Our intent was to report findings with statistically significant p-values, not to highlight their importance as key drivers in our analysis. We have revised the manuscript to better reflect this context and to avoid giving the impression of overstating a mechanistic relationship.

In line 177, the phrase “hormone and ER/PR/HER2 receptor status” should be revised. "hormone status" refers to ER and PR.

  • Thank you for your feedback. We have revised the phrasing to “hormone receptor and HER2 receptor status” for clarity and accuracy.

Reviewer 2 Report

Comments and Suggestions for Authors

 Strengths

  • This study highlights the consistent predictive value of the Breast Cancer Index (BCI) for extended endocrine therapy across racially diverse groups. By including underrepresented populations and applying robust statistical analyses, the findings support BCI’s utility in guiding personalized treatment decisions for hormone receptor-positive breast cancer in multiethnic clinical settings.

Constructive general comments

  • Confounding Factors Not Fully Addressed
    The interpretation of lower recurrence risk among Japanese and other Asian/Pacific Islander patients needs a more nuanced discussion of potential confounders such as socioeconomic status, treatment adherence, insurance coverage, and access to care.

  • Absence of Key Populations
    Black and Hispanic patients—who often experience disparities in breast cancer outcomes—are not represented. The lack of data from these groups should be acknowledged as a key limitation.

  • Limited Clinical Detail
    Additional clinical variables (e.g., menopausal status, duration of prior endocrine therapy) could improve the depth of analysis. Please clarify whether such data were unavailable or excluded by design.

  • Handling of Missing Data
    The manuscript should explicitly state how missing data were addressed in the analysis (e.g., listwise deletion, imputation).

  • Figure
    Including graphs summarizing BCI scores by race/ethnicity and odds ratios for key predictors would enhance data visualization and interpretability.

Constructive specific comments

  • Small Sample Size and Uneven Distribution

    • Only 159 patients total, with some groups (e.g., Filipino, Other) comprising less than 15% of the cohort. This limits the power of subgroup comparisons and generalizability of race-based conclusions.

  • Potential Misinterpretation of Protective Factors

    • The study reports lower recurrence risk in Japanese and Other Asian/Pacific Islander women but does not sufficiently address confounding variables such as access to care, treatment adherence, and socioeconomic status.

  • Overreliance on BCI Alone

    • While BCI is useful, complementary clinical and pathological factors (e.g., menopausal status, duration of initial endocrine therapy) should be included or better emphasized.

  • Lack of Inclusion and Exclusion Criteria
    The manuscript does not clearly define the inclusion and exclusion criteria for the study population. Please specify the criteria used to select participants, including any exclusions based on clinical or demographic characteristics.
  • Unjustified Cutoffs for Recurrence Risk Categories (Page 2, Line 59)
    The classification of distant recurrence risk as low (<5%), mid (5–10%), or high (≥10%) is presented without any supporting explanation or reference. The rationale behind these specific thresholds should be explained and, if based on prior literature, appropriately cited.

  • Lack of Justification for Predictor Categorization (Page 2, Lines 67–69)
    The manuscript introduces specific cutoff values for tumor size, grade, and receptor status in the regression models (e.g., tumor size categories of 0.1–0.9 cm, 1–1.9 cm, 2+ cm; ER/PR status: 0–49%, 50–100%) without providing a justification or referencing prior studies. Please clarify how these categories were determined.

  • Supplementary Table 1,2 and 3 Classification Rationale
    The classification criteria used in Supplementary Table S1, 2 and 3 for BCI recurrence risk and recurrence scores lack explanation. The manuscript should include a justification for these classifications within the methodology or results section, along with appropriate references if these categories are based on established standards.

Author Response

Confounding Factors Not Fully Addressed
 The interpretation of lower recurrence risk among Japanese and other Asian/Pacific Islander patients needs a more nuanced discussion of potential confounders such as socioeconomic status, treatment adherence, insurance coverage, and access to care.

  • Thank you for highlighting this important issue. We fully agree that socioeconomic status, treatment adherence, insurance coverage, and access to care can significantly impact breast cancer outcomes. Unfortunately, our dataset did not include these variables, which we now explicitly acknowledge as a limitation in our Discussion

Absence of Key Populations
 Black and Hispanic patients—who often experience disparities in breast cancer outcomes—are not represented. The lack of data from these groups should be acknowledged as a key limitation.

  • We appreciate this observation. The absence of Black and Hispanic patients, groups known to experience breast cancer outcome disparities, is indeed a limitation of our study. We have emphasized this more clearly in the Discussion.

Limited Clinical Detail
 Additional clinical variables (e.g., menopausal status, duration of prior endocrine therapy) could improve the depth of analysis. Please clarify whether such data were unavailable or excluded by design.

  • These data were not routinely captured in the chart records available to us and were therefore not included in our analysis. This limitation has now been clarified in the Discussion

Handling of Missing Data
 The manuscript should explicitly state how missing data were addressed in the analysis (e.g., listwise deletion, imputation).

  • Missing data in our analysis required for regression analysis, such as Oncotype Dx recurrence scores, were handled using listwise deletion. We have explicitly stated this methodology in the revised Methods

Figure

Including graphs summarizing BCI scores by race/ethnicity and odds ratios for key predictors would enhance data visualization and interpretability.

  • We appreciate your recommendation. We agree that including visual summaries of BCI scores by race/ethnicity and odds ratios would enhance readability. We have now added appropriate graphs in the manuscript to address this suggestion.

Small Sample Size and Uneven Distribution

Only 159 patients total, with some groups (e.g., Filipino, Other) comprising less than 15% of the cohort. This limits the power of subgroup comparisons and generalizability of race-based conclusions.

  • We acknowledge that certain racial subgroups were underrepresented, limiting statistical power for subgroup comparisons. However, the strength of our study lies in its systematic evaluation of a real-world, racially diverse patient dataset, providing insight into groups that have been historically underrepresented in literature. We have now more clearly stated this limitation in the Discussion and emphasize the need for larger, more representative studies to validate and extend these findings.

 Potential Misinterpretation of Protective Factors

The study reports lower recurrence risk in Japanese and Other Asian/Pacific Islander women but does not sufficiently address confounding variables such as access to care, treatment adherence, and socioeconomic status.

  • We recognize the potential for misinterpretation of the lower recurrence risk observed among Japanese and Other Asian/Pacific Islander groups due to unmeasured confounding factors, such as socioeconomic status and access to care, which were not routinely available in the chart records. We now clearly state in the Discussion that these findings require further investigation.

Overreliance on BCI Alone

While BCI is useful, complementary clinical and pathological factors (e.g., menopausal status, duration of initial endocrine therapy) should be included or better emphasized.

  • Thank you for highlighting this. While BCI provides valuable information, additional clinical and pathological details were unavailable in our dataset. However, our data offer important insight into the association between race/ethnicity and BCI scores, helping to inform treatment decisions and explore the specific role of racial and ethnic factors. We explicitly acknowledge this limitation in the Discussion and emphasize the need to include these variables in future research.

 Lack of Inclusion and Exclusion Criteria
 The manuscript does not clearly define the inclusion and exclusion criteria for the study population. Please specify the criteria used to select participants, including any exclusions based on clinical or demographic characteristics.

  • We have revised the Materials and Methods section to explicitly define our inclusion criteria (early-stage HR+ breast cancer, availability of BCI scores, and reported race) and exclusion criteria (metastatic or triple-negative breast cancer, absence of BCI score, or missing race data). A total of 159 patients met these criteria and were included in our study.

Unjustified Cutoffs for Recurrence Risk Categories (Page 2, Line 59)
 The classification of distant recurrence risk as low (<5%), mid (5–10%), or high (≥10%) is presented without any supporting explanation or reference. The rationale behind these specific thresholds should be explained and, if based on prior literature, appropriately cited.

  • Thank you for your thoughtful comment. While we categorized BCI scores into low, intermediate, and high risk groups, there are no formally established thresholds in the literature that define specific percentages for distant recurrence risk. The <5%, 5–10%, and ≥10% categories were selected to provide a more interpretable framework for analyzing and presenting recurrence risk in our dataset. We have now clarified this in the Materials and Methods

Lack of Justification for Predictor Categorization (Page 2, Lines 67–69)
 The manuscript introduces specific cutoff values for tumor size, grade, and receptor status in the regression models (e.g., tumor size categories of 0.1–0.9 cm, 1–1.9 cm, 2+ cm; ER/PR status: 0–49%, 50–100%) without providing a justification or referencing prior studies. Please clarify how these categories were determined.

  • We selected specific cutoff points for tumor size, grade, and receptor status based on clinical relevance and natural distribution patterns within our dataset. When possible, cutoff points reflected whole numbers or midpoints in clinical practice (e.g., 50% ER/PR). In cases with small cell sizes, categories were collapsed to ensure model convergence. This rationale has been added to the Materials and Methods.

Supplementary Table 1,2 and 3 Classification Rationale
 The classification criteria used in Supplementary Table S1, 2 and 3 for BCI recurrence risk and recurrence scores lack explanation. The manuscript should include a justification for these classifications within the methodology or results section, along with appropriate references if these categories are based on established standards.

  • As stated earlier, BCI recurrence risk scores were classified into low, intermediate, and high-risk groups by our selection, as there are no widely accepted thresholds in the literature. The thresholds were chosen to provide a more interpretable framework for analyzing and presenting recurrence risk in our dataset. As for recurrence scores, Oncotype DX recurrence scores are defined as low-risk (<18), intermediate-risk (18−30), and high-risk (>30). In our study, we based our classification on the natural distribution patterns within our dataset. We have added these explanations in the Materials and Methods section to clarify how BCI recurrence risk and prediction scores were categorized in the tables.

Reviewer 3 Report

Comments and Suggestions for Authors

 The main manuscript should contain results obtained during the study not in an supplement.

The results should be presented in the main manuscript.

  • In defining the white race, I propose the term caucasian. It is the most commonly used term in PubMed for the white race.
  • I suggest placing the results described in the main manuscript in the table. This will make it easier for the readers to analyze them. Please consider whether moving the table from the supplements to the manuscript would be better to present data than descriptive.
  • The manuscript title suggests using genomic tests to predict extended endocrine therapy benefits and recurrence risks in diverse breast cancer patient populations. What genes were tested? and what kind of test was used?
  • Extended endocrine therapy (EET) beyond five years may reduce distant recurrence. In early-stage hormone-positive (HR+) breast cancer. This study evaluates such differences in ethnic or racial influence on the recurrence of this type of tumor. Racial/ethic differences in EET benefit prediction were not statistically relevant. What is new in this study?
  • As rightly noted, this manuscript is a short communication that indicates Japanese and other Asian/Pacific Islanders patients had significantly lower odds of higher recurrence risk compared to caucasian type patients, but studies on a larger group of patients would be more significant.
  • The Breast Cancer Index (BCI) test risk of recurrence and extended endocrine therapy is indicated for use in women diagnosed with hormone receptor-positive (HR+), lymph node-negative (LN- or LN positive 1-3 early stage invasive breast cancer, who are distant recurrence free. This test provides a quantitative estimate of the risk for both late post-5-year diagnosis, distant recurrence, and the cumulative distant recurrence risk over 10 years in patients treated with adjuvant endocrine therapy or adjuvant chemoendocrine therapy. This test was developed and its performance characteristics determined by Biotheranostics Inc.1998.
  • In line 114 is described that the study shows that BCI can become impactful in clinical decision-making to EET (it is well described in PubMed), results of this study show no significant differences when considering racial and ethnic differences in clinical decision-making.

Author Response

The main manuscript should contain results obtained during the study not in an supplement. The results should be presented in the main manuscript.

  • Thank you for your recommendation. We have moved the supplementary tables into the main manuscript text to enhance clarity and ease of interpretation.

In defining the white race, I propose the term caucasian. It is the most commonly used term in PubMed for the white race.

  • We appreciate this suggestion. We have updated the manuscript to consistently use the term "Caucasian" when referring to the White population

I suggest placing the results described in the main manuscript in the table. This will make it easier for the readers to analyze them. Please consider whether moving the table from the supplements to the manuscript would be better to present data than descriptive.

  • Thank you for your suggestion. In addition to placing the supplementary tables into the body text, we have also added graphs summarizing BCI scores by race/ethnicity and odds ratios for key predictors in the Results

The manuscript title suggests using genomic tests to predict extended endocrine therapy benefits and recurrence risks in diverse breast cancer patient populations. What genes were tested? and what kind of test was used?

  • We used the Breast Cancer Index (BCI), which evaluates the expression of 11 specific genes (including HOXB13 and IL17BR among others) to predict late distant recurrence risk and benefit from extended endocrine therapy.

Extended endocrine therapy (EET) beyond five years may reduce distant recurrence. In early-stage hormone-positive (HR+) breast cancer. This study evaluates such differences in ethnic or racial influence on the recurrence of this type of tumor. Racial/ethic differences in EET benefit prediction were not statistically relevant. What is new in this study?

  • Our study specifically contributes novel insights by evaluating racial and ethnic differences in BCI results, an aspect not previously well-explored. Although we found no significant racial/ethnic differences in the prediction of benefit from extended endocrine therapy, we identified significant differences in recurrence risk scores among specific subgroups compared to Caucasian patients. These findings help expand the understanding of racial and ethnic implications in genomic testing for breast cancer.

As rightly noted, this manuscript is a short communication that indicates Japanese and other Asian/Pacific Islanders patients had significantly lower odds of higher recurrence risk compared to caucasian type patients, but studies on a larger group of patients would be more significant.

  • We agree fully with your observation. While our findings indicate a lower recurrence risk in Japanese and other Asian/Pacific Islander groups compared to Caucasian patients, we acknowledge that larger studies are essential to confirm and further explore these observations. We have more clearly emphasized this in the Discussion.  

The Breast Cancer Index (BCI) test risk of recurrence and extended endocrine therapy is indicated for use in women diagnosed with hormone receptor-positive (HR+), lymph node-negative (LN- or LN positive 1-3 early stage invasive breast cancer, who are distant recurrence free. This test provides a quantitative estimate of the risk for both late post-5-year diagnosis, distant recurrence, and the cumulative distant recurrence risk over 10 years in patients treated with adjuvant endocrine therapy or adjuvant chemoendocrine therapy. This test was developed and its performance characteristics determined by Biotheranostics Inc.1998.

  • Thank you for this informative detail. We have ensured that our description of the Breast Cancer Index in the Introduction clearly reflects its established indications and clinical utility, with appropriate citations from the literature.

In line 114 is described that the study shows that BCI can become impactful in clinical decision-making to EET (it is well described in PubMed), results of this study show no significant differences when considering racial and ethnic differences in clinical decision-making.

  • Thank you for your observation. Our intent was to examine whether this impactfulness varies across racial and ethnic groups. Given that we observed no significant racial or ethnic differences in predicting benefit from EET, our findings support BCI's universal clinical applicability. We have further clarified this point in the Discussion.